# Controllable Attribute-Guided Image Generation with Causal Modeling

## Abstract

Attribute-guided generative models enable explicit control over image content through labeled attributes. However, they often struggle to disentangle individual attributes and mitigate undesired correlations among them. For example, adding eyeglasses may unintentionally alter a person's perceived age, as eyeglasses are often correlated with older individuals in the training data. In this work, we propose a novel attribute-guided generative framework designed to address these challenges. Our method learns a mask-based representation for each attribute label, encouraging disentanglement by limiting each attribute's influence to a small subset of representation dimensions while preserving the information necessary to represent the corresponding label. For attributes that exhibit inherent dependencies, we further introduce a causal conditioning strategy that explicitly models their causal relationships, enabling more faithful and controllable attribute manipulation. Extensive experiments on a wide range of datasets demonstrate the effectiveness of our framework in enhancing attribute-level controllability.

## 1 Introduction

Recent advances in generative modeling have sparked growing interest in controllable image generation, enabling applications such as photo editing, creative content design, and data augmentation. Unlike traditional generative models that generate images solely from random noise, controllable models leverage semantic conditioning signals to guide the generation process, producing outputs that better align with user intent.

Conditional generative models achieve such control by conditioning image synthesis on auxiliary inputs, including class labels (Karras et al., 2020; 2022; Dhariwal & Nichol, 2021; Peebles & Xie, 2023), textual prompts (Rombach et al., 2022; Esser et al., 2024; Labs, 2024b; Betker et al., 2023; Ramesh et al., 2022; Saharia et al., 2022), and structural signals such as depth maps (Zhang et al., 2023; Zhao et al., 2023). Among these approaches, text-to-image models have demonstrated remarkable success in generating realistic and diverse images from natural language descriptions.

Despite their flexibility, natural language prompts can be inherently ambiguous. Descriptions such as "slightly smiling" or "partially bald" may be interpreted differently across prompts or models, limiting reliability in applications that require precise and repeatable control (see Table 2 and Fig. 4). In this context, attribute-guided image generation offers a complementary approach by employing explicit semantic labels to specify desired visual characteristics. Such structured supervision enables more precise, interpretable, and measurable control over individual attributes, making it particularly appealing for applications that require fine-grained image manipulation.

Nonetheless, attribute-guided generative modeling faces two key challenges: **attribute disentanglement** and **mitigating unwanted attribute interference caused by correlated attributes**.

Firstly, given attribute labels, a key challenge is how to ensure the generative model learns disentangled representations for individual attributes. It is essential to ensure that modifications of one attribute do not unintentionally influence others; for instance, changing a `heel` attribute should not inadvertently alter other attributes like shoe's color (see Fig. 5). Although recent advancements in Generative Adversarial Networks

(GANs) (Hou et al., 2024; Dobler et al., 2022; Zhang et al., 2024), diffusion models (Yu et al., 2024; Peebles & Xie, 2023; Ma et al., 2024), and autoregressive models (Tian et al., 2024; Sun et al., 2024; Wu et al., 2024) have exhibited strong generative performances, these methods are primarily class-conditioned and are not explicitly optimized for fine-grained attribute disentanglement. Consequently, they offer limited support for precise, interpretable control over individual attributes.

Secondly, attribute labels often exhibit inherent correlations within training datasets. For instance, attributes such as `eyeglasses` and `age` are frequently correlated, making independent manipulation challenging. Prior works, like InterfaceGAN (Shen et al., 2020), attempt to mitigate such correlations by projecting attribute vectors onto orthogonal directions. However, this approach presupposes independence between attribute subspaces and relies on linear classifiers, which may inaccurately reflect complex attribute correlations and inadvertently result in unwanted modifications (Chen et al., 2022) (see first column in Fig. 3).

These observations raise a fundamental question: *How can generative models learn disentangled attribute representations while mitigating the influence of correlations among attributes, thereby enabling precise and robust attribute control?*

**Our Contribution.** To address these challenges, we propose a novel attribute-guided generative framework that combines a mask-based representation learning mechanism for attribute disentanglement with a causal conditioning mechanism for modeling correlations among attributes.

To promote attribute disentanglement, we introduce a mask-based representation learning approach that restricts the influence of each attribute to a small subset of representation dimensions while preserving the information necessary to encode the corresponding label. This encourages independent attributes to be represented separately, reducing unintended interactions during manipulation.

To mitigate the unwanted influence of correlated attributes during generation and manipulation, we further leverage causal discovery methods to infer the causal structure of the label space. Based on the inferred graph, we introduce a causal conditioning mechanism that conditions each attribute only on its causal parents. By incorporating causal structure into the conditioning process, our framework better accounts for attribute dependencies and improves attribute-level controllability.

## 2 Related Work

**Attribute-guided Generation and Controllable Generation.** Attribute-guided image generation is a key task in controllable image synthesis. Early methods build on conditional GANs (Mirza, 2014), generating images conditioned on class or attribute labels. Augmentation-aware GANs like DiffAug (Zhao et al., 2020), AugGAN (Hou et al., 2024), and ANDA (Zhang et al., 2024) improve training robustness by incorporating data augmentations and reducing label leakage. Latent-based approaches guide attribute control by manipulating latent representations. Some methods generate latents from attribute labels (Li et al., 2023; Suwała et al., 2024; Nie et al., 2021), while others learn attribute directions in latent space (Shen et al., 2020; Wu et al., 2021; Ling et al., 2021). InterfaceGAN (Shen et al., 2020), for instance, uses linear SVM boundaries for manipulation. In diffusion-based models, WPlus (Li et al., 2024) incorporates StyleGAN latents into the sampling process, while ConceptSlider (Gandikota et al., 2024) uses paired data to train LoRA-based adapters (Hu et al., 2022). Other methods enable attribute control via text prompts (Patashnik et al., 2021; Wei et al., 2023; Huang et al., 2023; Zhu et al., 2025).

**Causality-Aware Generative Models.** A growing body of work has incorporated causal reasoning into generative modeling. Early efforts such as CausalGAN (Kocaoglu et al., 2017) explicitly model label dependencies through causal graphs by generating labels according to a predefined causal structure. However, their approach requires training auxiliary GANs to model the full label distribution, which becomes increasingly challenging as the number of attributes grows. In contrast, our method leverages causal structure solely to guide representation learning and conditioning, avoiding the need to explicitly generate label distributions. A detailed comparison with CausalGAN is provided in Appendix D. Subsequent works have extended this idea to different generative frameworks. For example, CausalVAE (Yang et al., 2021) introduces a causal layer to capture causal relationships among latent variables, while CGNN (Goudet et al., 2018) jointly learns

causal graphs and structural functions by matching generated and observed distributions. Other approaches focus on causal data generation and causal graph learning, including GAN-based methods for causal discovery (Moraffah et al., 2020), tabular data generation (Wen et al., 2022; Van Breugel et al., 2021), and normalizing-flow-based causal modeling (Javaloy et al., 2024). In the image domain, causality has also been explored for controllable and counterfactual generation. CGN (Sauer & Geiger, 2021) decomposes image generation into multiple causal factors, such as shape, texture, and background, while CAGE (Bose et al., 2022) leverages causal effects to generate counterfactual images. In contrast to these works, our method focuses on attribute-guided image generation, where we exploit causal structure among attribute labels to improve attribute-level controllability and reduce unwanted interactions among correlated attributes.

## 3 Controllable Attribute-Guided Image Generation with Causal Modeling

Given a set of images $\mathbf{x}$ and their corresponding attribute labels $\{\mathbf{T}_i\}_{i=1}^{m}$, our goal is to learn a generative model that produces high-fidelity images faithfully aligned with the specified attributes. Notably, attribute labels can be readily obtained from diverse sources, including pretrained attribute classifiers, multimodal large language models (Hurst et al., 2024; Yang et al., 2024), and learned from weakly supervised annotations when explicit attribute labels are unavailable (Appendix C), making the proposed setting broadly applicable in practice. An overview of the proposed framework is shown in Fig. 1.

To achieve this goal, we need to address two fundamental challenges:

**Problem 1: Learning disentangled representations.** To effectively model and manipulate multiple attributes, it is essential to learn disentangled representations for each attribute that isolate the underlying latent factors of variation specific to their corresponding labels. Without proper disentanglement, representations can become entangled, resulting in unintended changes to unrelated features. For example, modifying the shoe type from flat to heels may also change the color, as illustrated in Fig. 5 (first two rows).

**Problem 2: Mitigating unintended influence from correlated attributes.** Attribute labels in real-world datasets often exhibit causal or spurious correlations. These dependencies can cause changes in one attribute to inadvertently influence others. For example, the attributes **age** and **eyeglasses** are commonly correlated—older individuals are more likely to wear glasses. Consequently, setting the **eyeglasses** attribute from 0 to 1 may unintentionally alter the apparent age in the generated image (see Fig. 2(a)). This entanglement poses a major challenge for fine-grained, controllable image generation.

### 3.1 Learning Disentangled Representations via Masking

To address both challenges—disentangling representations and mitigating unwanted correlations—it is essential to reconsider how attribute labels are incorporated into generative models. In particular, we begin by analyzing why standard approaches to conditional generation often fail to produce properly disentangled representations.

A common approach to attribute-guided generation treats attribute labels as class indicators and trains a class-conditioned generative model. For example, given a random noise vector $\epsilon$ and an attribute label vector $\mathbf{T}$, one can train a conditional GAN (Goodfellow et al., 2014) using the formulation $x = G(\epsilon, \mathbf{T})$, where $G$ is the generator. However, such models tend to learn entangled representations of the attributes in $\mathbf{T}$. As a result, modifying a single attribute may lead to unintended changes in other attributes (see the first row of Fig. 5).

**Data generating process.** To address these limitations, we aim to learn a separate latent representation for each attribute $\mathbf{T}_i$. Without proper constraints, many models can fit the data distribution but still produce entangled representations. To reduce this ambiguity, we assume the data is generated according to the following process:

$$\begin{aligned}
\mathbf{z}_i &:= \mathbf{f}_i^1(\epsilon_i)\mathbf{T}_i + \mathbf{f}_i^0(\epsilon_i)(1 - \mathbf{T}_i); \\
\mathbf{x} &:= g(\mathbf{z}_c, \mathbf{z}_1, \mathbf{z}_2, \ldots, \mathbf{z}_m),
\end{aligned} \tag{1}$$

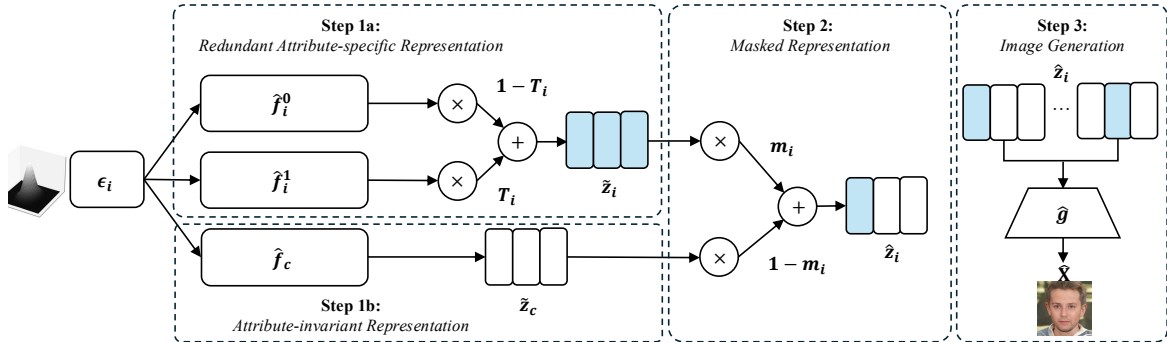

Figure 1: Overview of our mask-based framework for attribute-guided image generation (see Section 3.1 for details). Given noise samples $\epsilon$ from a prior distribution $\mathcal{N}(0, I)$, the model generates an attribute-invariant representation $\tilde{\mathbf{z}}_c$ and attribute-specific representations $\tilde{\mathbf{z}}_i$, which may contain redundant information. A learnable mask $\mathbf{m}_i$ is applied to modulate the contribution of each attribute $\mathbf{T}_i$, ensuring minimal influence from the attribute label $\mathbf{T}_i$ to inputs. All masked representations $\hat{z}_i$ are combined and passed into the generator $\hat{g}$ to synthesize the output image $\hat{\mathbf{x}}$.

where $\epsilon_i$ is sampled from a prior distribution such as $\mathcal{N}(0, I)$, , and $\mathbf{f}_i^1$, $\mathbf{f}_i^0$ are transformation functions associated with the $i$-th label. In this formulation, $\mathbf{z}_c$ represents the attribute-invariant component shared across all labels (e.g., the orientation of a human face). Each $\mathbf{z}_i \in \mathbb{R}^{d_i}$ is the latent representation for attribute $\mathbf{T}_i \in \{0, 1\}$, with dimensionality $d_i$. For simplicity, we denote the complete latent code as $\mathbf{z} = (\mathbf{z}_c, \mathbf{z}_1, \mathbf{z}_2, \ldots, \mathbf{z}_m)$. With this formulation, each attribute label $\mathbf{T}_i$ is responsible for a distinct subset of representation dimensions. Furthermore, these subsets are conditionally independent given the attribute labels, thereby promoting disentanglement.

**Estimation model.** Informed by the underlying data generation process, we first construct an **attribute-invariant representation** corresponding to the true latent factor $\mathbf{z}_c$ from Eq. 1:

$$\tilde{\mathbf{z}}_c = \hat{\mathbf{f}}_c(\epsilon_i), \tag{2}$$

where $\hat{\mathbf{f}}_c$ is a shared function across all attribute labels. As a result, the output $\tilde{\mathbf{z}}_c$ carries no information about any specific attribute label $\mathbf{T}_i$ (step 1b in Fig. 1). $\tilde{z}_c$ is intended to capture the information that is shared across images, e.g., image brightness.

Next, we construct an **initial redundant attribute-specific representation** for each label $\mathbf{T}_i$:

$$\tilde{\mathbf{z}}_i = \hat{\mathbf{f}}_i^1(\epsilon_i)\mathbf{T}_i + \hat{\mathbf{f}}_i^0(\epsilon_i)(1 - \mathbf{T}_i), \tag{3}$$

where $\tilde{\mathbf{z}}_i$ captures information specific to the attribute $\mathbf{T}_i$ (step 1a in Fig. 1). However, the amount of information each attribute conveys may vary significantly—for instance, the attribute `age` is likely to encode more complex features than `smile`. Moreover, if $\tilde{\mathbf{z}}_i$ has high dimensionality, it may inadvertently influence the representations learning of other attributes, such as $\tilde{\mathbf{z}}_j$ for $\mathbf{T}_j$, leading to undesired changes when only $\mathbf{T}_i$ is intended to be modified.

To address this, we propose a **mask-based estimation model**. Specifically, we introduce a learnable mask $\mathbf{m}_i$ for each attribute $\mathbf{T}_i$ and compute the final representation for attribute $\mathbf{T}_i$ as:

$$\hat{\mathbf{z}}_i = \mathbf{m}_i \odot \tilde{\mathbf{z}}_i + (1 - \mathbf{m}_i) \odot \tilde{\mathbf{z}}_c, \tag{4}$$

where $\odot$ denotes element-wise multiplication (step 2 in Fig. 1). This formulation enables us to control the degree to which each attribute affects the representation. For example, if $\mathbf{m}_i = \mathbf{1}$, then $\hat{\mathbf{z}}_i = \tilde{\mathbf{z}}_i$, meaning all elements of $\hat{\mathbf{z}}_i$ are influenced by attribute $\mathbf{T}_i$. Conversely, if $\mathbf{m}_i = \mathbf{0}$, then $\hat{\mathbf{z}}_i = \tilde{\mathbf{z}}_c$, which is attribute-invariant and ensures that modifying $\mathbf{T}_i$ has no effect on the output.

Finally, as shown in the final step of Fig. 1, we input all the masked representations into a generator $\tilde{g}$ to produce the output image $\hat{\mathbf{x}}$:

$$\hat{\mathbf{x}} = \tilde{g}(\tilde{\mathbf{z}}_c, \tilde{\mathbf{z}}_1, \tilde{\mathbf{z}}_2, \ldots, \tilde{\mathbf{z}}_m) \tag{5}$$

$$= \hat{g}(\hat{\mathbf{z}}_1, \hat{\mathbf{z}}_2, \ldots, \hat{\mathbf{z}}_m). \tag{6}$$

Following the conditional GAN framework, we employ a discriminator $D$ and perform adversarial training between the generator and discriminator to match the data distribution as:

$$\mathcal{L}_{\text{adv}} = \mathbb{E}_{(x,T)\sim p_{\text{data}}} \left[ \log D(x, T) \right] + \tag{7}$$

$$\mathbb{E}_{z\sim p(z),\, T\sim p(T)} \left[ \log \left( 1 - D(\hat{x}, T) \right) \right]. \tag{8}$$

To prevent each attribute from exerting excessive influence, we apply $\ell_1$ sparsity regularization to the masks, encouraging each to affect only a small number of representation dimensions:

$$\mathcal{L}_{\text{sparsity}} = \sum_{i=1}^{m} \|\mathbf{m}_i\|_1. \tag{9}$$

**Full Objective.** Our overall objective for training the attribute-guided generative model is:

$$\mathcal{L}_{\text{full}} = \mathcal{L}_{\text{adv}} + \lambda_{\text{sparsity}} \mathcal{L}_{\text{sparsity}}, \tag{10}$$

where $\lambda_{\text{sparsity}}$ is a hyperparameter that balances the adversarial loss and the sparsity regularization.

### 3.2 Mitigating Unintended Influence from Correlated Attributes through Causal Modeling

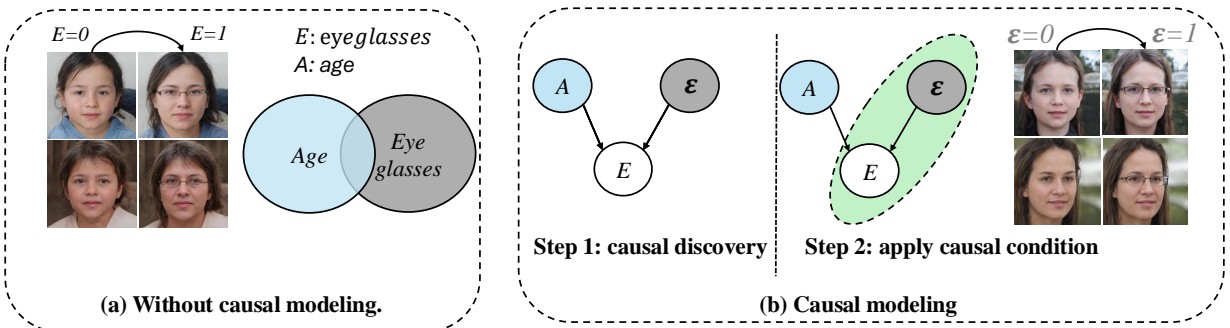

Figure 2: **The Necessity of Causal Modeling.** Modifying eyeglasses ($E$) without accounting for the causal relationship with age ($A$) can unintentionally change age-related features. By first discovering the causal direction between $A$ and $E$, and then applying causal conditioning, we can add eyeglasses without affecting perceived age (see Section 3.2).

While our mask-based estimation model enables learning disentangled representations with independent attribute labels where $\mathbf{T_i} \perp \mathbf{T}_j \ \forall_{i,j}$, in practice, these labels are often *causally related*, which can lead to unintended side effects when modifying a single attribute. For example, although the attributes age $A$ and eyeglasses $E$ are causally linked, we may wish to modify eyeglasses without altering the perceived age. Without explicit causal modeling, such changes can inadvertently affect age, as illustrated on the left side of Fig. 2.

This problem arises because age (denoted $A$) is a *parent node* of eyeglasses (denoted $E$). According to causal reasoning principles (Pearl, 2009), intervening on a child node (e.g., modifying $E$) should not influence its parent $A$. Violating this principle often results in unnatural or implausible generations. Therefore, it is crucial to incorporate causal reasoning into the generative process—particularly when both precise attribute control and natural image generation are desired.

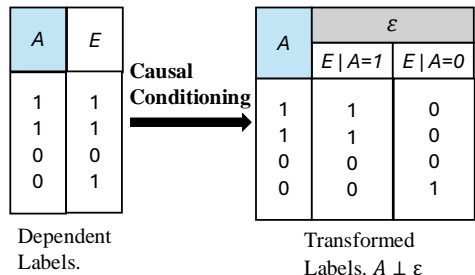

Table 1: **Example of Causal Conditioning. Example of Causal Conditioning.** After determining the causal relationship $A \rightarrow E$, we replace the original attribute label $E$ with a two-dimensional causally conditioned label $[E \mid A = 0, \ E \mid A = 1]$. The two components act as surrogates for the unobserved exogenous noise variable $\varepsilon$, capturing variations in $E$ that are independent of its parent attribute $A$. This representation enables the model to manipulate $E$ while reducing unintended changes to $A$. For example, let $A$ denote age and $E$ denote eyeglasses. In conventional attribute-guided generation, changing $E$ from 0 to 1 may inadvertently alter the perceived age due to correlations in the training data. In contrast, our method replaces $E$ with the causally conditioned label $[E \mid A = 0, \ E \mid A = 1]$. To add eyeglasses while preserving age, we modify the corresponding conditioned label (e.g., $E \mid A = 1$) from 0 to 1, while keeping the age attribute $A$ unchanged. As a result, eyeglasses can be added without substantially affecting the age appearance.

To this end, we adopt a two-step strategy (right side of Fig. 2): first, we discover the causal structure among attributes; then, we modify the model's conditioning mechanism to respect this structure.

**Step 1: Discovering the causal structure and direction.** To ensure causally consistent generation, we first identify the causal relationships among attributes (e.g., determining whether $A \rightarrow E$ or $E \rightarrow A$). Such relationships may be specified using domain knowledge or inferred from data using existing causal discovery methods. In our experiments, we employ the PC algorithm (Spirtes et al., 2001) to infer the causal graph. When the discovered graph contains unoriented edges, we resolve the ambiguities using domain-specific background knowledge. Details of the causal discovery procedure are provided in Appendix E.

**Step 2: Causal conditioning.** Suppose we have determined a causal relationship $A \rightarrow E$, represented by the structural causal model

$$E := f_E(A, \varepsilon),$$

where $\varepsilon$ is an exogenous noise variable independent of $A$, and $f_E$ is an unknown deterministic function. This formulation implies that $A$ and $E$ are dependent due to their shared causal mechanism.

To modify $E$ in the generated output while preserving its parent attribute $A$, we would ideally intervene on the exogenous noise variable $\varepsilon$, since $\varepsilon$ directly influences $E$ and is independent of $A$. However, because $\varepsilon$ is unobserved, we instead characterize its effect through the conditional distributions $P(E \mid A = 0)$ and $P(E \mid A = 1)$. Under the structural causal model, once $A$ is fixed, any remaining variation in $E$ is attributable to $\varepsilon$. Consequently, these conditional distributions capture the influence of the unobserved noise and provide a practical surrogate for manipulating $E$ without altering $A$.

**Summary.** Based on this analysis, our causal modeling strategy proceeds as follows (illustrated in Fig. 2(b)). Given a set of attribute labels (e.g., left in Table 1), we first use causal discovery algorithms to infer the directionality of causal edges. Then, we replace each child node $E$ with its corresponding conditional variables $E|A = 0$ and $E|A = 1$. After applying the causal conditioning transformation, we obtain a new set of attribute labels that encode causal structure. We then train the conditional generative model using these transformed labels, following the approach described in Section.3.1.

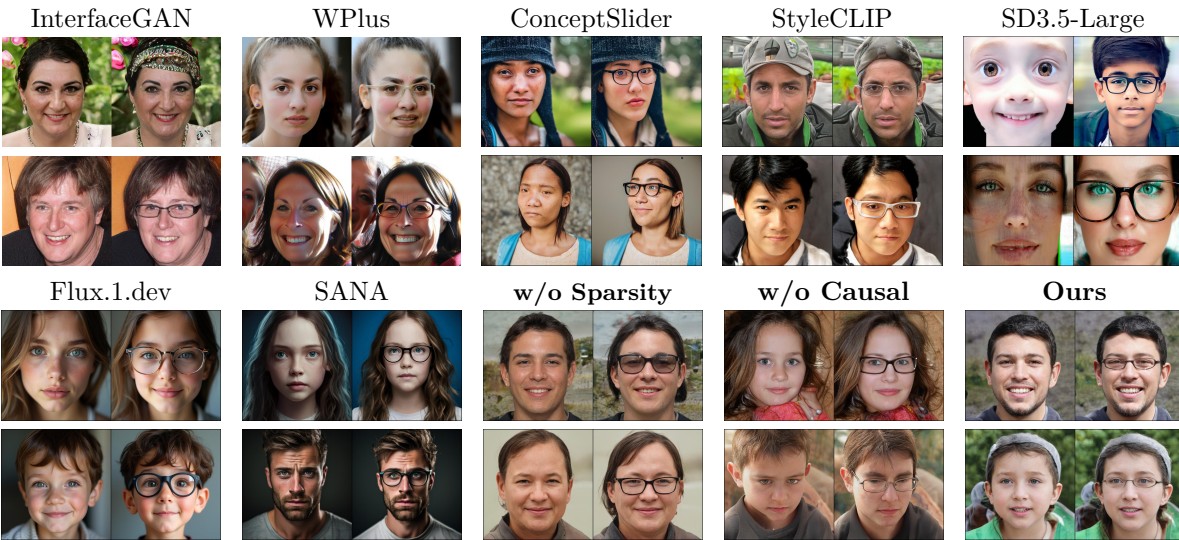

InterfaceGAN    WPlus    ConceptSlider    StyleCLIP    SD3.5-Large

Flux.1.dev    SANA    **w/o Sparsity**    **w/o Causal**    **Ours**

Figure 3: **Generation results after eyeglasses attribute modification.** Text-to-image (T2I) models such as SD3.5-Large (Esser et al., 2024), Flux.1.dev (Labs, 2024b), and SANA (Xie et al., 2024) often exhibit significant identity loss in the generated pairs, suggesting that textual prompts alone may be insufficient for precise attribute control. Attribute-guided methods preserve identity better but still suffer from distortions—for instance, InterfaceGAN (Shen et al., 2020) relies on a linear classifier to decorrelate attributes, which can be inaccurate and sometimes fails to add eyeglasses. In contrast, our method modifies only the target attribute while preserving other attributes, enabled by our mask-based representation learning and causal conditioning framework.

# 4 Experiments

In this section, we evaluate the proposed framework on controllable attribute-guided image generation tasks. We first describe the experimental setup, including implementation details, baselines, and evaluation metrics, in Section 4.1. We then present the main experimental results in Section 4.2, where we compare our method with state-of-the-art attribute-guided and text-to-image models on attribute manipulation tasks. Next, we investigate smooth attribute interpolation to assess fine-grained controllability. Finally, we conduct ablation studies to quantify the contributions of the proposed mask-based representation learning and causal conditioning mechanisms.

## 4.1 Setup

**Implementation, Baselines, and Metrics.** We develop our method based on the StyleGAN2-ADA code (Karras et al., 2020). We provide our code and implementation details in AppendixF. For causal-related attribute generation, we compare with InterfaceGAN (Shen et al., 2020), StyleCLIP (Patashnik et al., 2021), WPlus (Li et al., 2024), and ConceptSlider (Gandikota et al., 2024), SD3.5-Large (Esser et al., 2024), Flux.1.dev (Labs, 2024a), and SANA (Xie et al., 2024). We measure the performance with ArcFace ID (Deng et al., 2019) similarity between the generated paired faces, DINO (Caron et al., 2021) similarity, accuracy with pretrained classifier, and $L_1$ distance between the generated paired images. Following InterfaceGAN (Shen et al., 2020), we also perform a re-scoring disentanglement analysis, which evaluates how other attributes change when an edit is applied to a target attribute. We refer to this metric as Disen.

## 4.2 Experiment Results

**Attribute-guided Generation as a Complement to Text-to-Image Models** Recent text-to-image (T2I) models have demonstrated impressive performance. However, attribute-guided generation remains a valuable complement, particularly when precise and smooth generation or editing is required. As shown

| Method | Eyeglasses | | | | |
|---|---|---|---|---|---|
| | DINO ↑ | ID ↑ | Acc ↑ | $L_1$ ↓ | Disen ↓ |
| InterfaceGAN | 0.85 | 0.61 | 0.63 | 0.11 | 4.19 |
| StyleCLIP | 0.88 | 0.63 | 0.90 | 0.07 | 3.31 |
| WPlus | 0.85 | 0.77 | 0.91 | 0.09 | 4.16 |
| ConceptSlider | 0.80 | 0.74 | 0.92 | 0.06 | 2.82 |
| SD3.5-Large | 0.60 | 0.43 | **1.00** | 0.19 | 4.66 |
| Flux.1.dev | 0.72 | 0.48 | **1.00** | 0.12 | 3.98 |
| SANA | 0.78 | 0.49 | 0.97 | 0.11 | 3.26 |
| **Ours** | **0.95** | **0.78** | 0.96 | **0.04** | **2.53** |

| Attr | Method | DINO ↑ | ID ↑ | Acc ↑ | $L_1$ ↓ | Disen ↓ |
|---|---|---|---|---|---|---|
| | ConceptSlider | 0.85 | **0.73** | 0.70 | 0.08 | 3.04 |
| Beard | SANA | 0.83 | 0.39 | **1.00** | 0.13 | 4.49 |
| | **Ours** | **0.93** | 0.68 | **1.00** | **0.06** | **2.86** |
| | ConceptSlider | 0.93 | 0.78 | 0.75 | 0.06 | 2.39 |
| Chubby | SANA | 0.80 | 0.18 | 0.84 | 0.20 | 4.77 |
| | **Ours** | **0.96** | **0.79** | **1.00** | **0.06** | **1.96** |
| | ConceptSlider | 0.96 | 0.83 | 0.97 | **0.04** | 1.68 |
| Smile | SANA | 0.87 | 0.29 | **1.00** | 0.17 | 4.30 |
| | **Ours** | **0.98** | **0.90** | **1.00** | **0.04** | **1.57** |

Table 2: **Results on face attribute manipulation.** We compare our method with attribute-guided editing approaches and recent text-to-image models on four facial attributes. DINO and ID measure semantic and identity preservation between the original and edited images, Acc measures target attribute modification accuracy, $L_1$ measures the magnitude of unintended image changes, and Disen evaluates attribute disentanglement following InterfaceGAN (Shen et al., 2020). While recent T2I models achieve high attribute accuracy, they often introduce substantial changes to identity and other attributes. In contrast, our method consistently achieves strong attribute control while better preserving identity and reducing unintended attribute modifications.

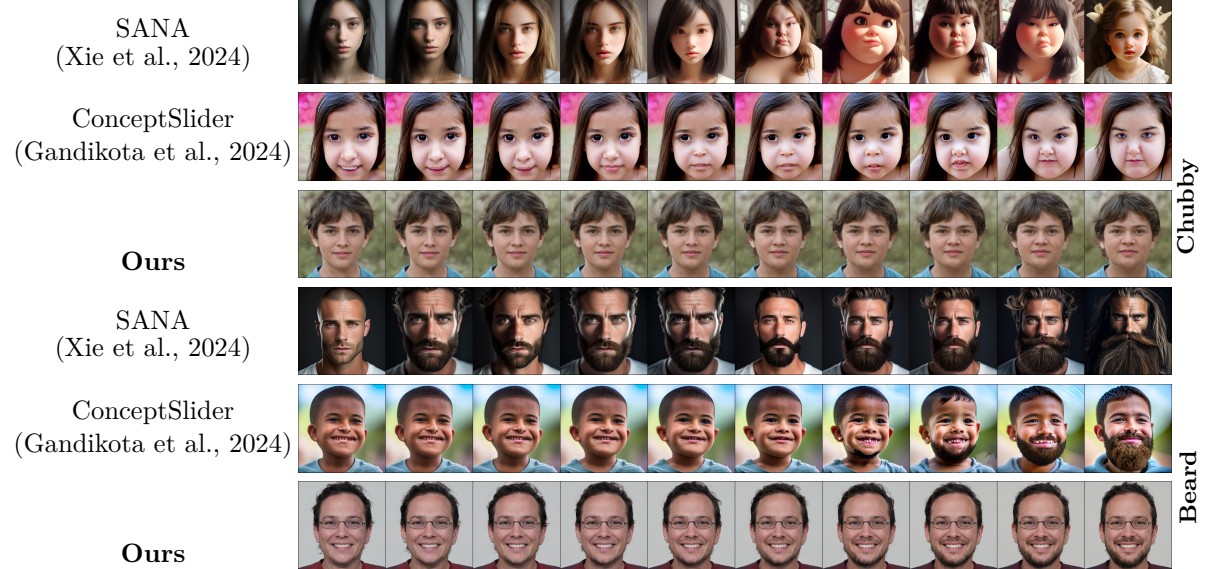

Figure 4: **Comparison of interpolations for attribute control.** The baseline method (Gandikota et al., 2024) exhibits a sudden-change phenomenon: at low control strengths, it introduces unintended modifications to other attributes, followed by abrupt activation of the target attribute. In contrast, our method achieves smooth and continuous interpolations, consistently modifying only the intended attribute.

in Table 2, state-of-the-art T2I models achieve strong instruction-following capabilities (evidenced by high target edit accuracy). Nevertheless, even minor modifications to the text prompt (e.g., a single word) can cause drastic changes in the output, as reflected in the low DINO, ID, L1, and Disen scores. Furthermore, we prompted GPT to enumerate 10 variations of facial size and hair and then generated corresponding images. As illustrated in Fig. 4, even subtle changes in the prompt often lead to identity loss and unsmooth interpolations.

**Comparisons with Baselines.** Quantitative results are reported in Table 2, and qualitative results are shown in Fig. 3. Our method achieves the best performance in attribute-controlled image generation, particularly in challenging cases with correlated attributes. For instance, the `eyeglasses` attribute is known to

| Method | Chubby CLIP↓ | Chubby DINO↓ | Beard CLIP↓ | Beard DINO↓ |
|---|---|---|---|---|
| ConceptSlider | 0.16 | 7.05 | 0.35 | 15.88 |
| SANA | 1.22 | 42.35 | 0.99 | 51.89 |
| SD3.5-Large | 1.72 | 119.67 | 1.53 | 90.99 |
| Flux.1-dev | 1.70 | 82.26 | 1.19 | 43.52 |
| **Ours** | **0.08** | **4.41** | **0.10** | **3.63** |

| Dataset | FID↓ Base Model | FID↓ $\lambda_{sparsity} = 0$ | FID↓ Ours |
|---|---|---|---|
| AFHQDog | 10.03 | 10.47 | **9.25** |
| ZAPPOS | 42.95 | 4.15 | **3.96** |
| ColorMNIST | 110.56 | 16.36 | **4.32** |
| LSUNBed | 28.06 | 17.75 | **8.09** |

Table 3: **Quantitative comparison of interpolation.** For each method, we linearly increased the target attribute value ((Gandikota et al., 2024) and Ours) or used progressively stronger prompts generated by GPT for T2I models to produce interpolated images. We then measured the accumulated DINO and CLIP differences over 10 sequential images (100 samples). Despite the close semantic similarity of the prompts, T2I models exhibited large accumulated differences between adjacent images, whereas attribute-based methods achieved smaller, smoother transitions. This indicates that for tasks requiring fine-grained, continuous control, attribute guidance can effectively complement T2I models.

Table 4: **Quantitative ablation study on the proposed masked representations.** Causal conditioning is not applied since attributes are independent. Compared to the base model that takes the entire attribute vector as input (Karras et al., 2020), our approach learns modular representations for each attribute, resulting in more stable outcomes. Incorporating the sparsity constraint further improves performance, demonstrating the effectiveness of the learned mask representation.

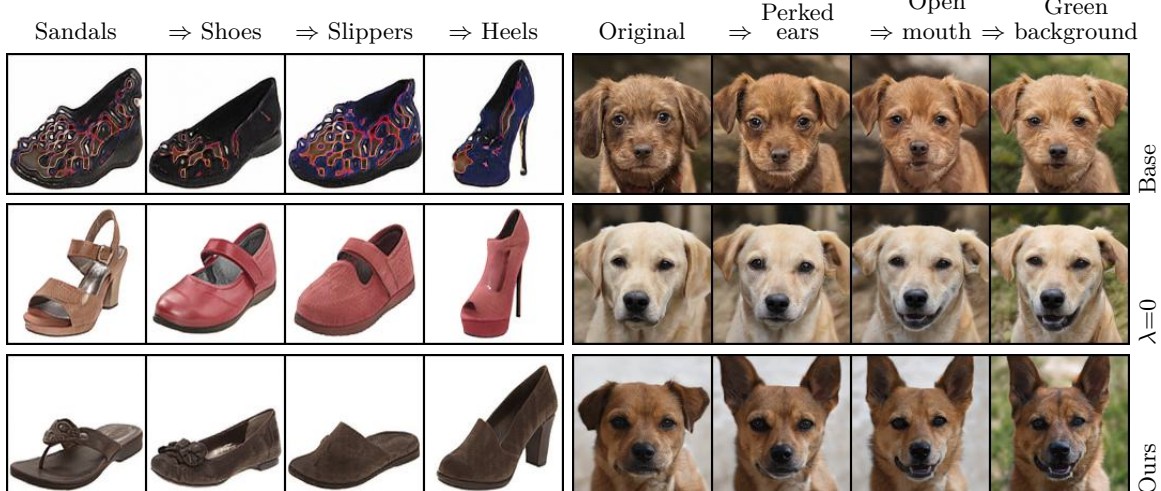

Figure 5: **Qualitative ablation study of the proposed masked representations.** The base model, StyleGAN2-ADA, does not learn separate representations for individual attributes and therefore entangles multiple attributes within a shared latent space. As a result, modifying one attribute often causes unintended changes to others. For example, changing the shoe color also alters other shoe characteristics. When our method is trained without the sparsity constraint ($\lambda = 0$), attribute interference is reduced but still observable, leading to less accurate attribute manipulation. In contrast, the proposed mask-based representation successfully isolates attribute-specific information, enabling the target attributes to be modified while preserving unrelated visual characteristics.

be strongly correlated with `age` (Shen et al., 2020). InterfaceGAN (Shen et al., 2020) attempts to mitigate this issue using orthogonal projection, but the linear boundary obtained by its classifier is often inaccurate, leading to unsatisfactory results (see Fig. 3). More recent methods such as WPlus (Li et al., 2024) and ConceptSlider (Gandikota et al., 2024), built on the Stable Diffusion v1.4 backbone, improve upon InterfaceGAN but still fail to account for attribute correlations. As a result, modifying one attribute (e.g., eyeglasses) often causes unintended changes in others, such as age or pose.

**Smooth Interpolation.** To comprehensively evaluate attribute controllability, we perform interpolation experiments. Specifically, we gradually increase the target attribute value or prompt GPT to generate ten semantically ordered descriptions for the attribute. We then measure the accumulated CLIP and DINO embedding differences across ten interpolated images (100 samples). As shown in Table 3, attribute-guided methods—ConceptSlider (Gandikota et al., 2024) and ours—produce significantly smaller accumulated differences than text-to-image (T2I) models. This suggests that prompt-based approaches are less effective for fine-grained attribute control, highlighting the necessity of explicit attribute guidance.

Qualitative results are shown in Fig. 4. The strong baseline ConceptSlider (Gandikota et al., 2024) performs interpolation by adjusting its LoRA scale. However, it often suffers from abrupt transitions—either introducing irrelevant modifications at low control strengths or suddenly activating the target attribute along with unwanted changes. In contrast, our method applies the target attribute smoothly and consistently as the control value increases, without influencing other attributes. These results demonstrate the effectiveness of our disentangled attribute representation in enabling stable and continuous control.

**Ablation Study.** In this paper, to achieve faithful and controllable attribute-guided generation, we introduce two key components: (1) a mask-based representation to isolate attribute influence, and (2) causal conditioning to mitigate correlations among attributes.

We first evaluate the effectiveness of the proposed mask-based representation for attribute-guided generation. Specifically, we compare three configurations—our base model (StyleGAN2-ADA (Karras et al., 2020)), our model without the sparsity constraint ($\lambda_{\text{sparsity}} = 0$), and our full model—across four datasets: AFHQDog, ZAPPOS, ColorMNIST, and LSUNBed. Since the attributes in these datasets are independent, causal conditioning is not applied. As shown in Table 4, the mask-based representation enables the model to learn modular, attribute-specific features, leading to more stable and disentangled generation compared to the base model, which directly uses the entire attribute vector. Incorporating the sparsity constraint further refines the learned representation, yielding a more compact and effective latent space. Qualitatively, Fig. 5 shows that with mask sparsity, attribute manipulations become cleaner and more disentangled—modifying one attribute no longer induces unwanted distortions in other visual factors.

We further analyze the impact of the sparsity constraint and causal modeling on face attribute editing (Fig. 3). Without the sparsity constraint, the `eyeglasses` attribute influences many unrelated regions, causing unintended visual changes (DINO: 0.935, ID: 0.730). Without causal modeling, sparsity alone maintains partial disentanglement, but modifying `eyeglasses` still introduces correlated artifacts, such as making the subject appear older (DINO: 0.943, ID: 0.734). In contrast, our full model (DINO: 0.953, ID: 0.775) achieves more robust disentanglement, enabling precise and isolated control over attributes while minimizing side effects.

## 5 Limitations, Discussion, and Conclusion

Although our method achieves strong performance across a variety of attribute-guided image generation tasks, it is currently designed for settings in which semantic attributes are available or can be reliably inferred. While such labels can often be obtained from pretrained classifiers, multimodal foundation models, or weakly supervised annotations, extending the framework to more open-ended forms of control remains an interesting direction for future work. In particular, integrating the proposed mask-based representations and causal conditioning mechanism with free-form textual prompts may enable a unified framework that combines the flexibility of language-based conditioning with the precision of attribute-level control.

In this paper, we proposed an attribute-guided image generation framework that addresses two key challenges: learning disentangled attribute representations and mitigating the unwanted influence of correlated attributes. Our approach is grounded in a data-generating formulation and introduces a mask-based representation learning mechanism that restricts each attribute's influence to a small subset of representation dimensions. To account for correlations among attributes, we further propose a causal conditioning strategy that leverages the inferred causal structure during generation. Extensive experiments across multiple datasets demonstrate that our method consistently improves attribute-level controllability and disentanglement while maintaining high image quality.

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

## A  Appendix Summary

This appendix provides additional details, analyses, and experimental results that complement the main paper:

- Section B presents additional generation and interpolation results, including comparisons with the strong baseline ConceptSlider (Gandikota et al., 2024).

- Section C discusses extensions of our framework to settings where attribute labels are unavailable.

- Section D compares our method with CausalGAN (Kocaoglu et al., 2017), including both theoretical analysis and empirical results.

- Section E presents the complete causal graph used in our experiments.

- Section F provides implementation details and training configurations. The source code is also included in the supplementary materials.

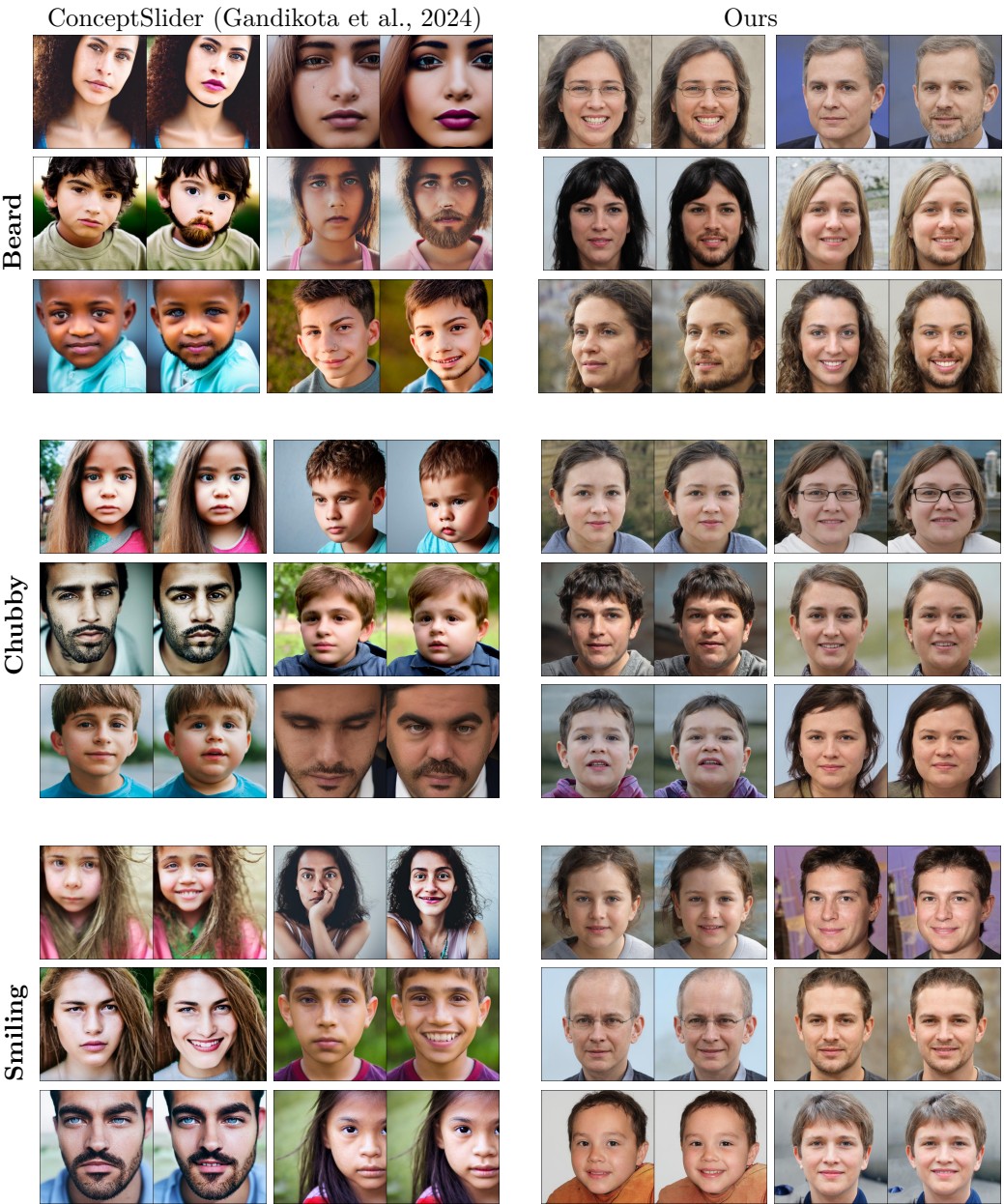

Figure 6: **Comparisons of Attribute-Guided Generation.** The baseline method, ConceptSlider (Gandikota et al., 2024), often fails to generate beards for uncommon attribute combinations, such as a girl with a beard. Additionally, we observe that ConceptSlider sometimes produces overly heavy beards or exaggerated smiles—even when using the same strength parameter—while in some cases it fails entirely. These inconsistencies suggest that the method is unreliable for controllable generation. In contrast, our approach accurately recovers the latent factors corresponding to the target attribute labels, enabling faithful modification of the desired attribute without affecting unrelated features.

## B  More Empirical Results

**Comparison with ConceptSlider**  We provide additional generated samples in Fig. 6. Interestingly, the baseline ConceptSlider (Gandikota et al., 2024) often fails to generate the intended attributes when faced with rare combinations. For example, it fails to add a beard to a female face, instead altering the lip color

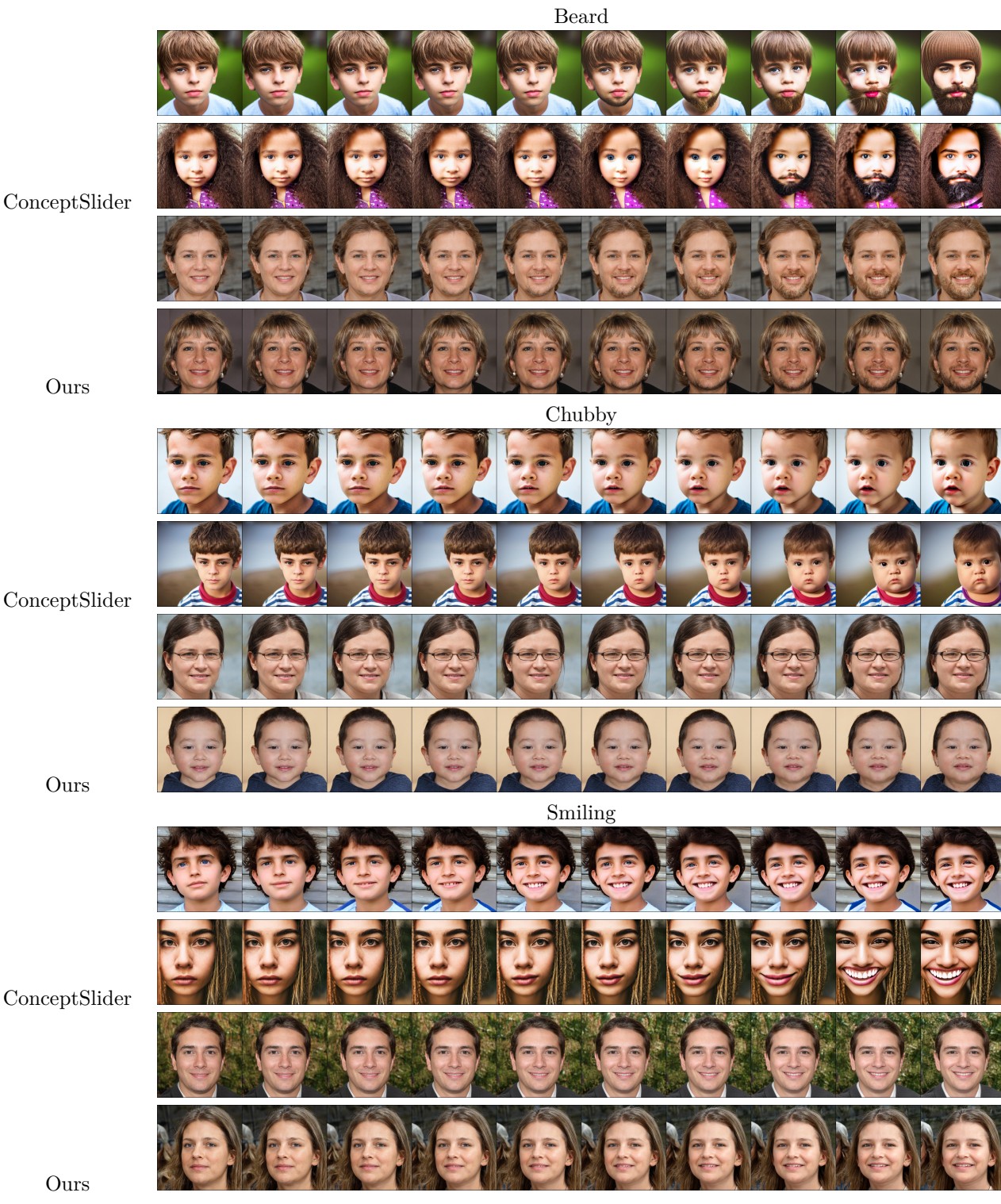

Figure 7: **Interpolation Comparisons.** The baseline method, ConceptSlider (Gandikota et al., 2024), exhibits a sudden-change phenomenon: at lower strength values, there is little to no effect, but beyond a certain threshold, it introduces abrupt and excessive changes—including alterations to unrelated attributes. In contrast, our method demonstrates more stable and controlled behavior, gradually modifying only the target attribute as the strength increases.

(first row). Moreover, when using the same strength parameter, ConceptSlider sometimes produces overly heavy beards or no beard at all, indicating that each sample may require its own carefully tuned hyperparameter. This inconsistency suggests that the baseline method is less reliable for controllable, attribute-guided generation. When attempting to generate chubby faces, it either changes the identity of the original subject (fifth row) or introduces irrelevant modifications, such as altering clothing color (fourth row). In contrast, our method successfully generates the target attributes—even in uncommon scenarios like beards on female faces—while preserving unrelated features and faithfully modifying only the intended attribute.

**Interpolation Comparison** We present the interpolated samples in Fig. 7. Similar to the results shown in the main paper, the baseline ConceptSlider (Gandikota et al., 2024) suffers from a sudden-change phenomenon, where the target attribute appears abruptly and is often accompanied by unintended modifications. For example, it suddenly adds a heavy beard to a female face while also altering the eyes (second row). In another case, it modifies unrelated attributes during interpolation—for instance, the clothing of the boy changes as the strength of the "smile" attribute increases (ninth row). In contrast, our method achieves smooth and gradual interpolation between the absence and presence of the target attribute, while preserving all other attribute values.

## C   Learning Without Attribute Labels

Our method, as introduced in the main paper, relies on access to attribute labels. With these labels, our model can disentangle the underlying concepts associated with each attribute. However, in real-world scenarios, such labels may not always be available. In this section, we propose solutions to address this limitation and demonstrate that our model can also *learn attribute labels jointly* when they are not provided.

First, we relax the requirement for detailed attribute labels and instead assume access to *class labels* only. We then introduce an auxiliary *labeler network* that takes the class label as input and predicts the corresponding attribute labels. This labeler network is trained jointly with our main model. To regularize the predictions and encourage sparsity—we apply an additional $\ell_1$ penalty to the estimated attribute labels. In other words, we expect the predicted attribute labels to be sparse.

We find that this simple design is effective on several datasets. As shown in Fig. 8, our method is capable of learning meaningful attribute labels, such as digit color and shape, directly from data and class labels. Furthermore, Fig. 9 demonstrates that our method can also discover distinct concepts in more challenging datasets—for example, learning both painting styles and subjects using only class labels and image data.

Second, we explore the most challenging setting, where only *image data* are available, and no labels (class or attribute) are provided. Since we have shown success in learning attribute labels with only class supervision, we propose extending our approach to the fully unsupervised case using *clustering techniques*. For example, (Liu et al., 2020) demonstrated that it is possible to cluster images during GAN training. We suggest adapting their method to cluster samples and estimate attribute labels in conjunction with our model. We leave this extension as a promising direction for future work.

## D   CausalGAN and Causal Conditioning

CausalGAN also assumes that there exists a causal relationship among the attribute labels. To capture this, it introduces an additional *causal controller* to model the joint distribution of the labels. Specifically, suppose a causal graph $X \rightarrow Q \leftarrow Y$. CausalGAN first generates $X$ and $Y$ using neural networks $x = f_x(z_x)$ and $y = f_y(z_y)$, respectively. It then generates $q = f_q(x, y, z_q)$, where $z_x$, $z_y$, and $z_q$ are latent variables. In other words, it requires using $n$ separate neural networks to model the causal generation process when there are $n$ attribute labels. Finally, all generated attribute labels are concatenated and fed into the discriminator to match the distribution of real labels.

**Causal Perspective.** CausalGAN (Kocaoglu et al., 2017) aims to model the full causal generative process of the attribute labels. This involves learning the causal relationships between all pairs of variables. For instance, the causal controller in CausalGAN can be used to measure the influence of ancestors on a

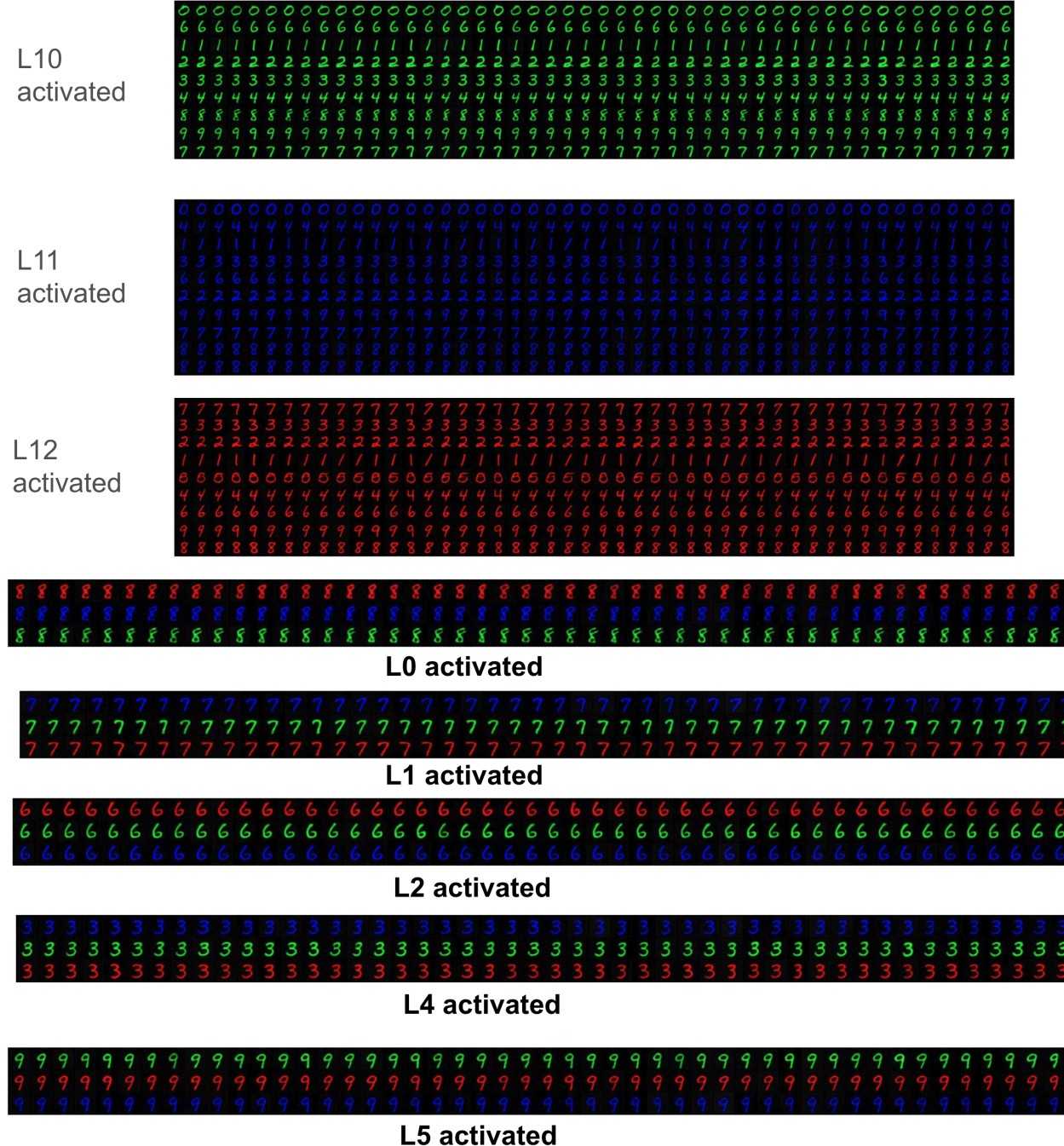

Figure 8: We can learn the attribute labels (denoted by $L$) jointly with our model **when attribute labels are not given**. On this MNIST dataset, we are able to learn the digit shape labels and digit colors from data.

$L_0$ activated

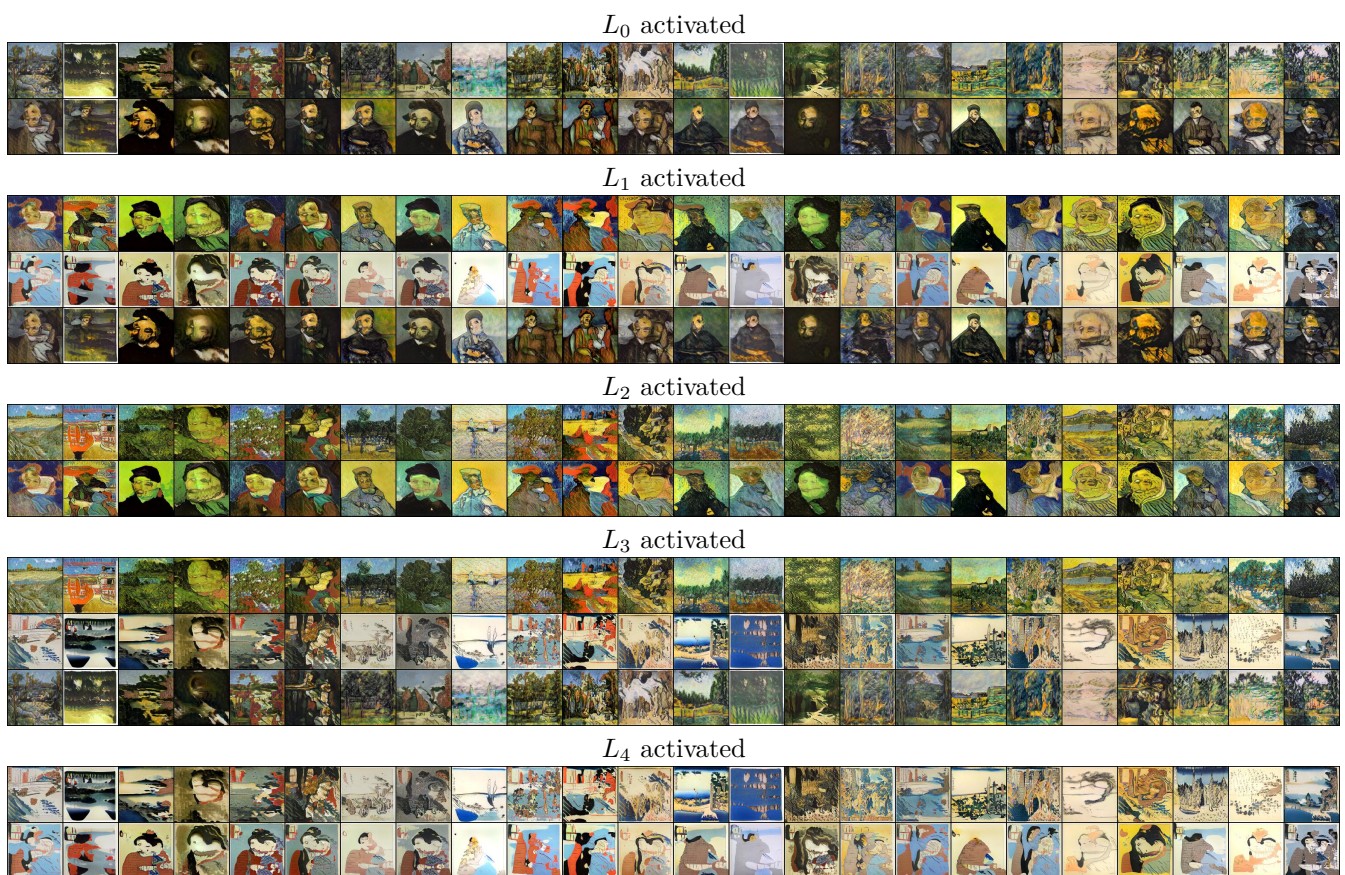

$L_1$ activated

$L_2$ activated

$L_3$ activated

$L_4$ activated

Figure 9: We can learn the attribute labels (denoted by $L$) jointly with our model **when attribute labels are not given**. On this artist dataset, we successfully disentangle different concepts, e.g., human face and painting style. For example, $L_1$ denotes the human face attribute and $L_2$ denotes the Van-Gogh painting style.

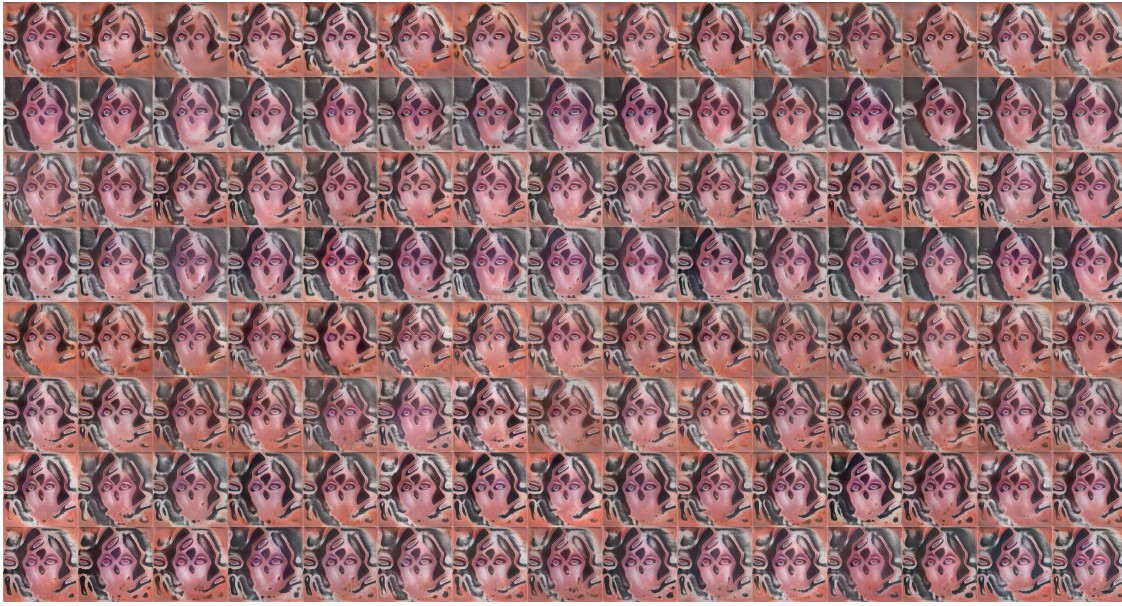

Figure 10: CausalGAN requires a large amount of training data to model complex causal relationships among a greater number of attribute labels. We train its causal controller on the FFHQ dataset, but it fails to fully match the distribution of attribute labels. Moreover, due to the randomness introduced by noise in the causal controller, it generates combinations of labels that never appear in the training data. As a result, CausalGAN quickly fails during the subsequent image generation training.

given node. However, such an approach demands large amounts of training data to accurately learn these dependencies—a point we validate through experiments.

In contrast, our method focuses only on modeling the causal relationships from parent nodes to a given attribute, which is sufficient for handling the correlations in the data. For example, generating eyeglasses may inadvertently increase the perceived age, even though the true causal direction is age → eyeglasses. Our causal conditioning approach is therefore simpler (requiring no separate pretraining) and more data-efficient.

**Empirical Comparison.** CausalGAN was originally implemented on a smaller causal graph with around 10 variables using the CelebA-64×64 dataset (162,770 images). For a fair comparison in our setting, we re-implemented CausalGAN using the StyleGAN architecture. However, we found that the 70,000 samples available in the FFHQ-512×512 dataset were insufficient to model the complex causal relationships among the 37 attribute labels.

We trained the causal controller for 100 epochs using the recommended WGAN-GP loss. By the end of training, the outputs of the causal controller appeared nearly binary, matching the format of the real data. However, due to the limited training data and stochastic noise introduced by the controller, it often generated combinations of attribute labels that never occurred in the training set. As a result, when these unrealistic labels were used to train the full CausalGAN, the discriminator easily distinguished between real and fake samples, causing training to collapse prematurely. As shown in Fig.10, it collapses quickly.

To fully compare our causal condition with the causal controller, we introduce another version of CausalGAN in our setting. Specifically, we still use real labels during training to avoid failure, i.e., the trained model will be StyleGAN2-ADA. After training, we apply the causal controller to see if we are able to sample some interventional images. And we term this new model as CausalGAN$^+$. We present the results in Fig. 11. We observe that CausalGAN$^+$ introduce unnecessary changes to the images when we only want to change a single attribute.

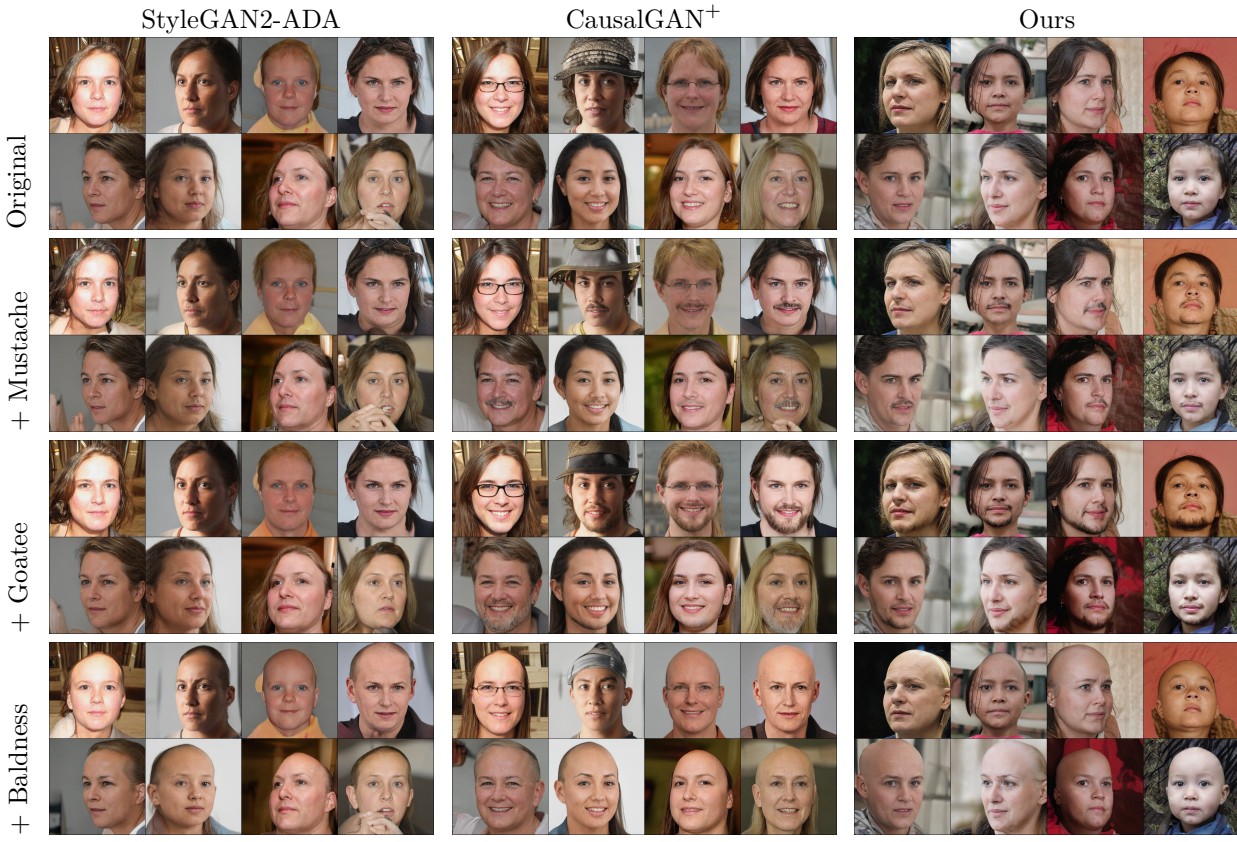

Figure 11: Comparison of controllable image generation. We first fix the random noise and generate girl images. Then we activate the mustache, goatee and baldness concepts for each method. The StyleGAN2-ADA is unable to generate female with mustache images. While the CausalGAN is able to add such attributes, we observe significant unnecessary changes. In contrast, our method is able to change the target attributes only.

## E  Causal Discovery

We perform causal discovery on the FFHQ attribute labels and the causal graph is shown in Figure 12. Specifically, we use the PC algorithm (Spirtes et al., 2001) and the BOSS algorithm (Andrews et al., 2023), the latter being designed to efficiently handle large numbers of variables and dense graphs. We first apply the PC algorithm to obtain an initial causal graph, then use BOSS to verify its structure. If both algorithms produce the same edge, we treat it as reliable. In practice, we find that PC outputs generally align well with human domain knowledge. While some edges remain unoriented, our goal is not to develop a new causal discovery algorithm, but simply to obtain enough structural information to guide controllable image generation. In ambiguous cases, we follow established practice in applied causal modeling by incorporating minimal domain knowledge to resolve uncertain directions.

## F  Implementation Details

We provide the training code in the supplementary material. Our method is built upon StyleGAN2-ADA (Karras et al., 2020), with our main empirical contribution focused on a redesign of the mapping network used in StyleGAN2-ADA. In traditional StyleGAN-based approaches, Gaussian noise and a class embedding are jointly processed by an MLP to produce an entangled latent representation in the $\mathcal{W}$ space.

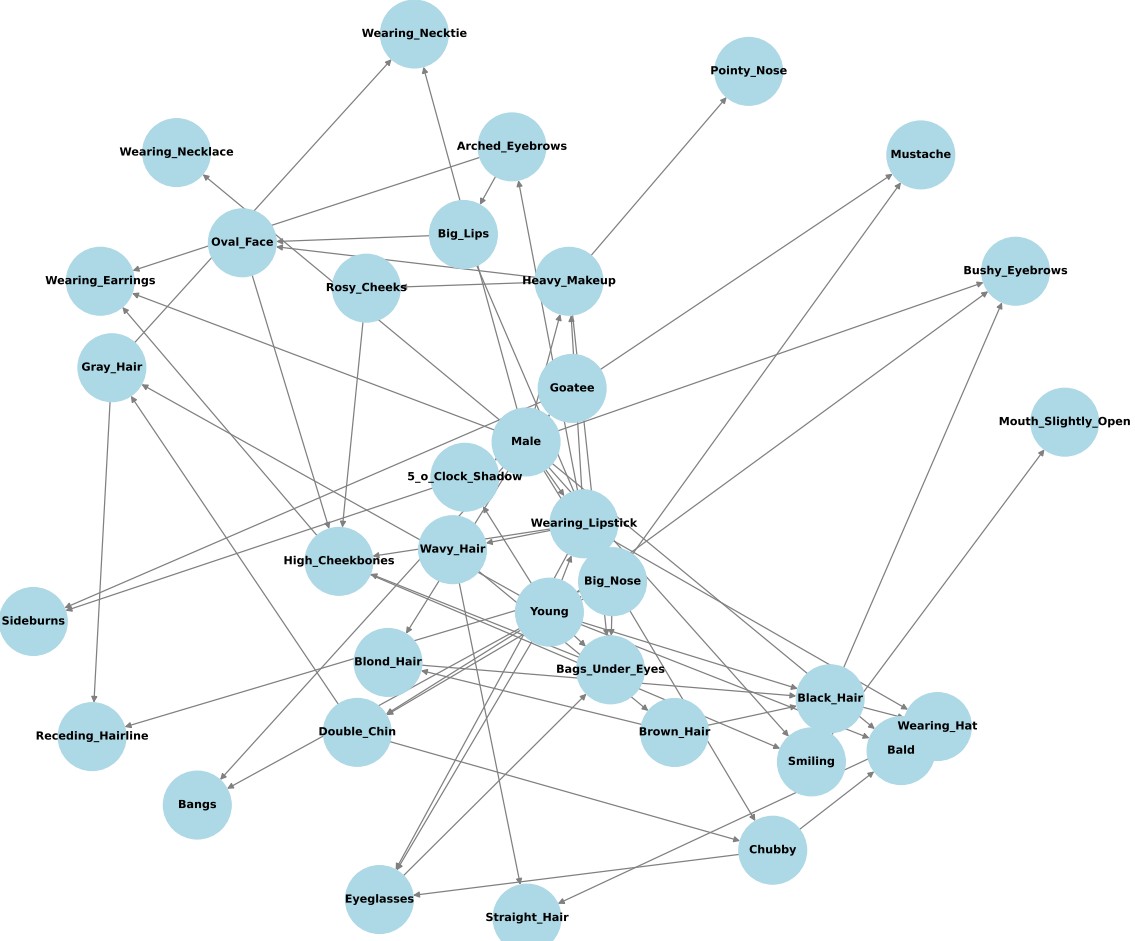

Figure 12: Causal discovery result on the FFHQ attributes labels.

Ours (w/o sparsity)      Ours (w/o causal)      Ours

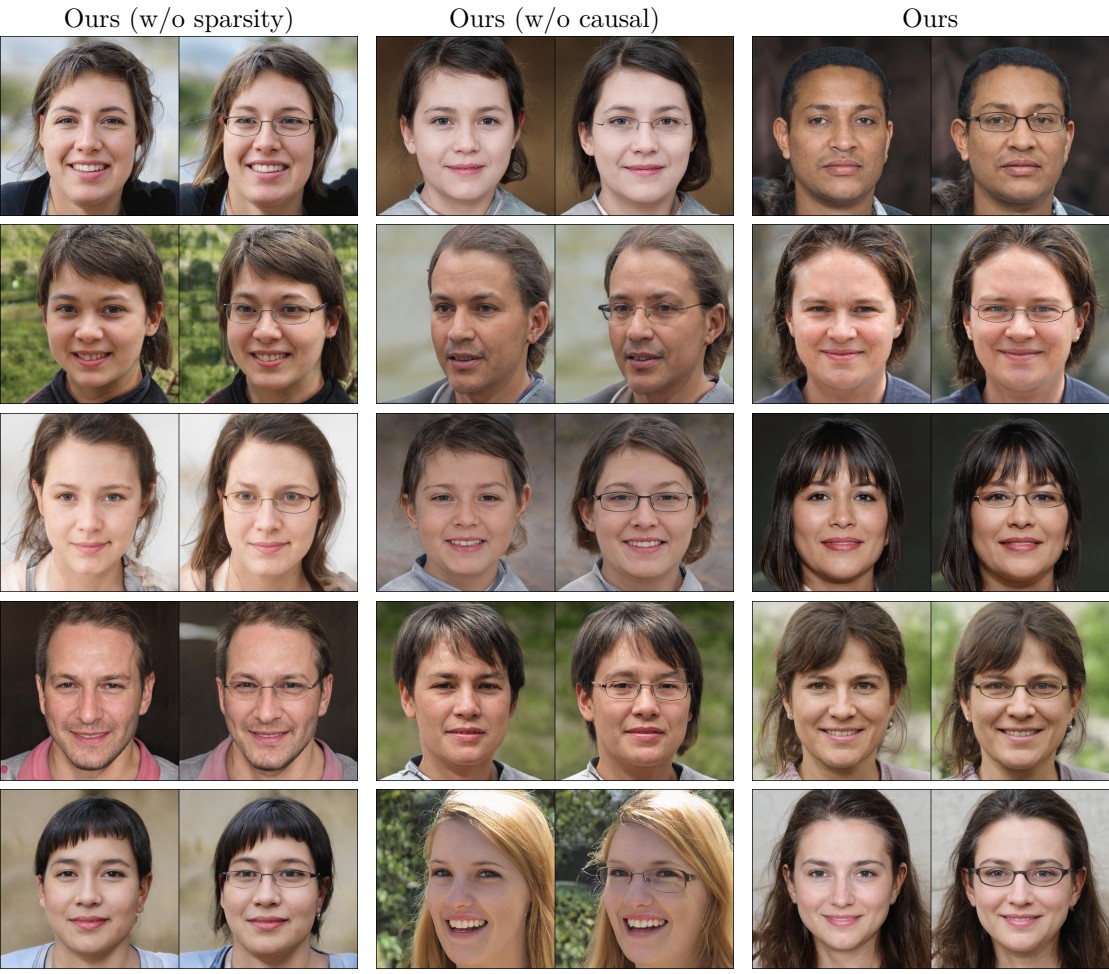

Figure 13: **Qualitative Ablation Results**. Without sparsity and causal modeling, the output images suffer more distortions when we want to add eyeglasses only.

For each attribute $A_i$, we employ a two-layer MLP to transform an input noise vector $\epsilon_i$ into an activated or deactivated concept representation. These outputs are then concatenated to form a latent vector **z**, yielding a new latent space $\mathcal{Z}$ instead of the conventional $\mathcal{W}$ space.

At the beginning of training, we set the dimensionality of each attribute representation to 20. Additionally, we introduce a learnable mask $m_i$ for each attribute to allow the model to select the relevant dimensions for each concept. To promote sparsity, we apply an $L_1$ regularization on the mask, typically using a sparsity weight of $\lambda_{\mathrm{sparsity}} = 0.1$.

