# OpenReview forum: "Controllable Attribute-Guided Image Generation with Causal Modeling"
_TMLR — Under review for TMLR_

### Review · Reviewer_owqy · 2026-07-01

**Summary Of Contributions:**

This manuscript studies attribute-level controllability: the problem when changing one attribute affects different attributes in attribute-guided image generation. Two methods, such as a learnable mask for latent attribute representation and causal discovery, are presented. Experiments based on StyleGAN2-ADA were performed.

**Audience:**

Yes

**Audience Explanation:**

Attribute-guided image generation would be important and several researchers may find certain values in this study.

**Claims And Evidence:**

No

**Claims Explanation:**

I think the current version is not sufficient. Please see the Requested Changes below.

**Requested Changes:**

- I suggest the authors disentangle causation and correlation. “Age causes eyeglasses” might seem plausible, but much stronger qualitative and quantitative evidence of causation should be provided.
- More controlled experiments are required for the causal conditioning, because the advantage could arise from other factors such as more fine-grained label representation. The exact behavior for the proposed causal conditioning is something like a parent-gated label channel, rather than a conditional distribution.
- The comparison should be fair; the proposed method is trained on the specific datasets, whereas the baselines are simply general-purpose models such as SD3.5-Large.
- Eq. 4 defines masked representation, but Eq. 5 uses an unmasked one. Is this intended?
- The adversarial loss L_{adv} should be a maximization objective. Nevertheless, the author used it in L_{full} for minimization.
- Is it correct to say z ~ p(z)? The term z would not be the latent representation that is independently sampled from the label.
- If input noise is shared, invariant noise across attributes, then it should be \epsilon_{c} rather than \epsion_i.
- Attribute T_i is said to be T_i \in {0, 1}, which is binary and is not compatible with categorical attributes in practice. It should be rewritten.
- Check typos:
    - “Example of Causal Conditioning. Example of Causal Conditioning.”
    - “, , and”
    - “AppendixF”
    - “Section.3.1”
    - “the representations learning” → “the representation learning”
    - “CausalGAN+ introduce” → “CausalGAN+ introduces”
    - “suffer more distortions” → “suffer from more distortions”
    - “Fig.10”
    - “For example, (Liu et al., 2020) demonstrated” → “For example, Liu et al.,( 2020) demonstrated”
    - Incorrect authors for “Conditional Generative Adversarial Nets”

---

### Review · Reviewer_Bond · 2026-07-09

**Summary Of Contributions:**

The paper tackles controllable attribute-guided image generation, targeting two well-known failure modes: (i) attribute entanglement, where editing one attribute (e.g., shoe type) unintentionally changes another (e.g., color), and (ii) unwanted interference from correlated attributes, where editing a child attribute (e.g., eyeglasses) drags along a correlated parent (e.g., age). The authors propose two components built on top of StyleGAN2-ADA:

Experiments cover face attributes (eyeglasses, beard, chubby, smile) on FFHQ and independent-attribute datasets (AFHQ-Dog, Zappos, ColorMNIST, LSUN-Bed). Baselines include latent-editing methods (InterfaceGAN, StyleCLIP, WPlus, ConceptSlider) and recent T2I models (SD3.5-Large, Flux.1-dev, SANA), evaluated with DINO/ArcFace-ID/attribute-classifier-accuracy/L1/re-scoring disentanglement, plus an interpolation-smoothness analysis.

**Additional Comments:**

I sincerely apologize for the delay in providing this review. I have carefully evaluated your work and have the following comments

**Audience:**

Yes

**Audience Explanation:**

Yes. Controllable/disentangled generation, latent-space attribute editing, and the intersection of causal modeling with generative models are all active topics well within TMLR's readership (the paper itself builds on TMLR-published work, e.g., Bose et al. 2022). The specific problem of correlated-attribute interference during editing is practically important and comparatively under-served by existing latent-direction methods, and the proposed causal-conditioning trick is simple enough to be adopted by practitioners. Even readers who do not work on generation would find the "stratify the child label by its parent as a surrogate for the exogenous noise" idea a clean, reusable observation. The breadth of baselines (including 2024–2025 T2I and slider methods) also makes the empirical comparison a useful reference point for the community

**Claims And Evidence:**

No

**Claims Explanation:**

### Strengths
- Clear, real problem; the correlated-attribute interference case is under-addressed by prior latent-editing work, and the motivation (Figs. 2–3, 5) is compelling.
- The causal-conditioning idea — stratifying a child label by its parent's value as a surrogate for exogenous noise — is simple, intuitive, and cheap to apply on top of a standard conditional GAN.

### Weakness
- The causal contribution—which represents half of the paper’s claimed novelty—is insufficiently supported by empirical evidence compared to the mask contribution. The current evaluation is limited to a single pair of values for one attribute, lacking a per-pair breakdown, directional ablation, and variance analysis.
- No seeds/error bars provided; the performance gaps are quite small (e.g., 0.943 vs. 0.953 for DINO). This doesn't necessarily imply that seeds/error bars are required, but the margins are narrow.

**Requested Changes:**

Qualitative comparisons do not use a shared source image, confounding the visual claims. In Figs. 3 and 4, each method is shown starting from a different identity (e.g., in Fig. 4 the "Chubby" rows begin from a girl for SANA/ConceptSlider but a boy for Ours; in Fig. 3 all eleven panels are different faces). Since the paper's qualitative claims are precisely about smoothness and identity/attribute preservation, these cannot be assessed when the starting images differ across methods. For the cross-backbone baselines this should be handled by inverting a single common source image into each model's latent space (standard practice) or at minimum matching identity. Critically, the ablation panels (w/o Sparsity, w/o Causal, Ours) share the same backbone and latent space, so there is no obstacle to using the identical seed/latent — as shown, they start from different faces, which makes the qualitative ablation uninterpretable. Please regenerate all qualitative comparisons (Figs. 3, 4, 6, 7) from a fixed set of shared source images, holding the seed constant within each comparison.

Moreover, the editing baselines (InterfaceGAN, StyleCLIP, ConceptSlider, WPlus) operate on a fixed source image by construction and use shared sources in their own papers, so a common-source comparison is the established norm here, not an unavoidable difficulty. The paradigm gap also affects the metrics: "Ours" forms its evaluation pair by flipping the attribute label under a shared noise ε, which structurally favors identity/L1 preservation, whereas the editing baselines must preserve identity through inversion-plus-edit. The authors should (a) build all qualitative comparisons from a common inverted source, and (b) clarify how the evaluation pairs are constructed for each method so that the ID/L1/DINO numbers in Table 2 are comparable.

---

### Review · Reviewer_ra4r · 2026-07-15

**Summary Of Contributions:**

This paper introduces an attribute-guided generative framework designed to solve two persistent issues in controllable image synthesis: attribute entanglement and spurious correlations.

To isolate individual attributes, the authors propose a mask-based representation learning mechanism that uses sparsity regularization to restrict each attribute's influence to a minimal subset of latent dimensions. Furthermore, to handle inherent dependencies in the training data (e.g., the correlation between age and eyeglasses), the framework employs a causal conditioning strategy. By inferring a structural causal graph, the model conditions attributes only on their causal parents, thereby mitigating unintended feature alterations during manipulation.


## Key Strengths

- The authors introduce a mask-based estimation model paired with an L1 sparsity constraint to force dimensional separation. Assuming the data generating process relies on conditionally independent subsets is a sound approach to minimizing cross-attribute leakage.

- The method leverages causal discovery algorithms (PC and BOSS) to explicitly model data structure. This represents a significant upgrade over prior works like InterfaceGAN, which naively assume independence between attribute subspaces and rely on linear classifiers that fail to capture complex correlations.

- The paper successfully demonstrates a vulnerability in current state-of-the-art text-to-image models, such as SD3.5-Large and Flux.1.dev. It proves that these models suffer from severe identity loss and abrupt transitions during fine-grained interpolation, so it justifies the continued need for explicit attribute guidance.


## Key Weaknesses

- To me, the biggest weakness is that **the framework's success is entirely bottlenecked by the accuracy of the causal discovery step.** The authors admit that algorithms like PC often yield unoriented edges, forcing them to rely on ad-hoc, domain-specific background knowledge to manually resolve ambiguities. If the assumed causal graph is incorrect, the conditioning mechanism will structurally enforce the wrong data distribution.

- The framework is implemented on top of the StyleGAN2-ADA backbone. While it performs well on highly structured datasets like FFHQ and AFHQDog, it is questionable whether this mask-based dimensional partitioning assumption can scale to the highly entangled latent spaces of modern diffusion models.

- The core methodology requires a predefined set of explicit semantic labels during training. While the authors propose a workaround using auxiliary labeler networks for weakly supervised settings, the framework fundamentally lacks the flexibility of natural language conditioning, limiting its applicability for open-ended or free-form generation.

**Audience:**

Yes

**Audience Explanation:**

The researchers working on image generation will be interested in this submission.

**Claims And Evidence:**

Yes

**Claims Explanation:**

Overall, the claims are supported by empirical evidence, but they come with significant caveats.

The evidence is only convincing within the narrow, highly curated scope of StyleGAN architectures and specific datasets like FFHQ. Furthermore, the manual intervention required to resolve ambiguous edges in the causal graph severely weakens the claim of a scalable solution.

**Requested Changes:**

N/A

---

### Review · Reviewer_bYy7 · 2026-07-20

**Summary Of Contributions:**

The authors propose a novel multi-attribute conditioned GAN for controlled image generation. Their proposed architectural modifications to the standard approach include (a) block-wise attribute representation disentanglement, (b) sparse masking of the dimensions in these block-wise attribute representations, and (c) a causal graph-informed transformation of the attributes to a new space that they argue can reduce interference during generation between otherwise causally-related attributes. They run extensive experiments to compare their method to several baselines across several metrics, and find that their model performs best.

**Audience:**

Yes

**Audience Explanation:**

It’s possible the TMLR audience will find this work interesting, but on the other hand I think that the paradigms and interests of the field have likely changed to diverge quite far from the current approach. The current focus right now is on conditioning information is specified in natural language (which is far richer than fixed attributes in virtue of its compositionality) where controlled edits are made over several iterations through natural language follow-up requests. I think the paradigm described here is perhaps too inflexible to be useful in practice. That being said, the authors do compare to current T2I methods and find some genuine benefits.

**Broader Impact Concerns:**

There are probably clear ethical concerns around the use of these methods for deep fakes that convincingly edit images of real people, but the authors don’t attempt to engage with these questions. I think this work can probably benefit from a broader impact statement, or at least some nods to ethical concerns surrounding their work.

**Claims And Evidence:**

No

**Claims Explanation:**

- I’m not clear on why the causal conditioning approach works. Really, exactly the same information is being used to train the model. Take Table 1 as an example. When $A=1$ it will always be the case that $(E|A=0)=0$, and when $A=0$ it will always be the case that $(E|A=1)=0$. The transformation therefore results in the conditioning variable having exactly the same information as without the transform, so where is the problem being solved? If $A$ was correlated with $E$ because it was a causal parent, well now it will just be correlated with the joint $[(E|A=0), (E|A=1)]$ instead, it seems to me. While this trick seems to work in practice, I don’t really see why in theory.
- The approach discussing causality ignores statistical correlations due to hidden confounders, which are going to be very common in practice (likely far more than direct causal connections).
- I have clarity concerns around the purpose of the mask. More details below around this point in my comments about requested changes.
- While I think that some of the architecture components are a bit ad hoc and the theoretical justifications are at times unclear, the results seem quite convincing and robust. A large number of baselines are considered (even T2I methods) with a large number of metrics, with reliable gains. The experiments are probably the stronger (and more convincing) part of this paper.

**Requested Changes:**

- I didn’t understand the justification for the masks, and their role in the architecture. The paragraphs explaining it seem a bit hand-wavy, and this will need to be clarified. As far as I can tell, since these masks are static per attribute, they just seem like they can at most assign different effective dimensionalities to each attribute’s latent representation (which the authors do mention, but along with some other independent arguments I didn’t understand). Additionally, it seems like a lot of information in $\tilde{\mathbb{z}}_c$ is going to essentially be replicated redundantly across a lot of the $\hat{\mathbb{z}}_i$’s.
- Table 2 doesn’t indicate which methods are T2I and which are not.
- “Furthermore, we prompted GPT to enumerate 10 variations of facial size and hair and then generated corresponding images. As illustrated in Fig. 4, even subtle changes in the prompt often lead to identity loss and unsmooth interpolations.” Fig. 4 doesn’t appear to have GPT, nor does it have facial size / hair interpolations. If what the authors meant was that they asked GPT to enumerate the 10 prompts, which they then fed to other T2I models, this isn’t clear from the text (in particular because GPT is now multi-modal).
- Small note: in Table 2, Chubby, ConceptSlider, the number should also be bolded (ties the proposed method).